# Efficiency Recycling and Utilization of Phosphate from Wastewater Using LDHs-Modified Biochar

**DOI:** 10.3390/ijerph20043051

**Published:** 2023-02-09

**Authors:** Chunxia Ding, Xiuyu Long, Guangyong Zeng, Yu Ouyang, Bowen Lei, Rongying Zeng, Jing Wang, Zhi Zhou

**Affiliations:** 1School of Chemistry and Materials Science, Hunan Agricultural University, Changsha 410128, China; 2College of Chemistry and Material Science, Hengyang Normal University, Hengyang 421001, China

**Keywords:** phosphate, LDHs, biochar, recycling, fertilizer

## Abstract

The excessive application of phosphate fertilizers easily causes water eutrophication. Phosphorus recovery by adsorption is regarded as an effective and simple intervention to control water bodies’ eutrophication. In this work, a series of new adsorbents, layered double hydroxides (LDHs)-modified biochar (BC) with different molar ratios of Mg^2+^ and Fe^3+^, were synthesized based on waste jute stalk and used for recycling phosphate from wastewater. The prepared LDHs-BC4 (the molar ratio of Mg/Fe is 4:1) has significantly high adsorption performance, and the recovery rate of phosphate is about 10 times higher than that of the pristine jute stalk BC. The maximum adsorption capacity of LDHs-BC4 for phosphate was 10.64 mg-P/g. The main mechanism of phosphate adsorption mainly includes electrostatic attraction, ion exchange, ligand exchange, and intragranular diffusion. Moreover, the phosphate-adsorbed LDHs-BC4 could promote mung bean growth, which indicated the recovery phosphate from wastewater could be used as a fertilizer.

## 1. Introduction

Phosphate is one of the main components of fertilizers and plays an important role during the growth process of crops [1,2]. However, with the massive application of phosphate fertilizers, it became the main cause of water eutrophication, which posed a variety of risks to the ecological environment [3]. Therefore, it is necessary to recovery phosphate from the phosphate wastewater.

Currently, phosphate recycling methods include chemical precipitation [4], adsorption [5], biological treatment [6], electrodialysis [7], and so on [8]. Among these methods, adsorption is widely used due to its simple operation. Common adsorbents such as activated carbon, zeolite, industrial resin, etc., have been used in adsorbing phosphate [9]. However, their low adsorption capacity limits their practical application [10,11,12]. Therefore, highly efficient adsorbents for phosphate should be developed.

Biochar (BC) has been widely used to purify wastewater and improve the soil [13,14,15,16,17], but pure BC has a relatively low adsorption capacity for phosphate because of the lack of special functional groups [18]. Therefore, it is necessary to modify BC and increase its special functional groups to improve the adsorption efficiency for phosphate. It was reported that layered double hydroxides (LDHs)-modified BC exhibited excellent adsorption performance for anion pollutants (e.g., phosphate, arsenate, chromate, and selenate) from wastewater [19,20,21]. The morphology of LDHs-BC includes a main layer composing of double metal hydroxides and an intercalation layer containing anions and water molecules, which display rich porosity and high anion exchange capacity [22,23]. For instance, BC was modified by LDHs composed of Mg^2+^, and Al^3+^ has richer special surface functional groups for phosphate adsorption [24]. Though there were some reports about the preparation of LDHs-BC and its application for phosphate-containing wastewater treatment, most of them focused on the adsorption mechanism of LDHs-BC. The effect of different molar ratios of the two metals in LDHs on the phosphate adsorption and whether phosphate-adsorbed LDHs-BC could be applied to crops have rarely been reported.

Thus, in this work, a series of LDHs-BC with different molar ratios of Mg^2+^ and Fe^3+^ (LDHs-BCx) were synthesized based on waste jute stalk. The influencing factors and mechanism of phosphate adsorption by LDHs-BCx were studied systematically. Furthermore, the desorption and bioavailability of P in the LDHs-BCx after adsorbing phosphate were also investigated. Finally, we tested the possibility of the adsorbed phosphate to be used as an agricultural phosphate fertilizer.

## 2. Reagents and Methods

### 2.1. Reagents

Waste jute (*Corchorus Capsularis* L.) stalk was obtained from experimental base of Hunan Agricultural University. MgCl_2_·6H_2_O, FeCl_3_·6H_2_O, KH_2_PO_4_, and NaOH were all of analytical reagent grade and were purchased from Sinopharm Chemical Reagent Co., Ltd., China.

### 2.2. Synthesis of LDHs-BC

Three LDHs-BCx composites with different molar ratios of Mg^2+^ and Fe^3+^ were prepared by coprecipitation method [25,26,27]. Firstly, 4.06 g MgCl_2_·6H_2_O was added into FeCl_3_·6H_2_O solution with different concentrations to the Mg/Fe molar ratio of 2, 3, and 4, respectively. Then, jute stalk powder (5 g) was added into each reaction flask containing Mg/Fe solution and vigorously agitated at 60 °C. The pH of the mixed solution was adjusted to 10 by 5 M NaOH and aged for 2 h at 60 °C. Afterward, the synthesized composites were washed with deionized (DI) water to achieve a neutral pH and were oven-dried. Finally, the dried composite was transferred into a tubular furnace and treated at 600 °C for 1 h. The as-synthesized composites by the method were named LDHs-BCx (where x is 2, 3 or 4).

### 2.3. Characterizations

Scanning electron microscopy was used to measure morphologies of the samples (SEM, Hitachi S-4800, Tokyo, Japan). The crystallinities and average grain size of LDHs-BCx were analyzed by XRD pattern at a scan speed of 5°/min from 5° to 80° (XRD, Shimadzu XRD-6000, Kyoto, Japan). The special functional groups of LDHs-BCx were determined by an FT-IR spectrometer (Bruker, Billerica, Massachusetts, USA) in the 400 to 4000 cm^−1^ range (Shimadzu Affinity-1, Kyoto, Japan). The specific surface areas and pore size of materials were measured by N_2_ adsorption-desorption at 77 K on a surface area and porosity analyzer (Quantachrome Quadrasorb SI, Graz, Austria). The surface chemical state of materials was analyzed by a X-ray photoelectron spectroscopy (XPS, Semmelvil, Waltham, MA, USA).

### 2.4. Adsorption Experiments

#### 2.4.1. Influence Factors

The adsorption capacity of LDHs-BCx was evaluated by batch experiments. The effects of initial phosphate concentration (10–60 mg-P/g), initial pH (3.0–11.0), contact time (0–24 h), different reaction temperatures (15–35 °C), and different anions (Cl^−^, SO_4_^2−^, CO_3_^2−^) on phosphate recovery were investigated. For every adsorption experiment, a certain amount of LDHs-BCx was added to the conical flask with 50 mL of 20 mg-P/L KH_2_PO_4_. Then the conical flask was put into a shaking table at 25 °C for 24 h at 150 rpm. Residual phosphate in the solution was measured with a UV-visible spectrophotometer, and calculation formula of adsorption capacity and removal rate were as follows [12]:(1)$$Q_{e}=\frac{\left(C_{0}-C_{e}\right)V}{m}$$
(2)$$R\left(\%\right)=\frac{\left(C_{0}-C_{e}\right)}{C_{0}}$$
where C_0_ and C_e_ are the initial and equilibrium concentrations (mg-P/L), respectively. V is the solution volume (L) and m is the adsorbent dosage (g).

#### 2.4.2. Kinetics, Isotherms, and Thermodynamics

Using different models (such as pseudo-first-order, pseudo-second-order, Elovich, and intraparticle diffusion) to fit the experimental data according to Equations (3)–(6) [28,29]:(3)$$q_{t}=q_{e}-q_{e}e^{-k_{1}t}$$
(4)$$q_{t}=\frac{k_{2}q_{e}^{2}t}{1+k_{2}{\mathrm{tq}}_{e}}$$
(5)$$q_{t}=\frac{1}{\beta }\ln\left(\alpha \beta t\right)$$
(6)$$q_{t}{=k}_{\mathrm{id}}t^{0.5}+C$$
where k_1_ and k_2_ are the equilibrium rate constant associated with the pseudo-first-order and pseudo-second-order, respectively. q_t_ is adsorption capacity of phosphate at t time. α is initial adsorption rate, and β is desorption constant. k_id_ is intraparticle diffusion rate constant, and C is determined by the thickness of the boundary layer.

Adsorption isotherms were also measured using different initial concentrations of phosphate at 25 °C for BC and LDHs-BCx. Experimental data were fitted by Langmuir, Freundlich, and Langmuir–Freundlich adsorption isotherm models according to Equations (7)–(9), respectively [30]:(7)$$q_{e}=\frac{q_{m}K_{L}\mathrm{Ce}}{1+K_{L}C_{e}}$$
(8)$$q_{e}=K_{F}C_{e}^{n}$$
(9)$$q_{e}=\frac{K_{\mathrm{LF}}q_{m}C_{e}^{n}}{1+K_{\mathrm{LF}}C_{e}^{n}}$$
where K_L_, K_LF_, and K_F_ represent Langmuir bonding term, Langmuir–Freundlich affinity coefficient, and Freundlich affinity parameter, respectively. q_m_ indicates the Langmuir maximum adsorption capacity and n is the Freundlich linearity constant.

Thermodynamic parameters were thought as important factors to provide information on the inherent energy changes in the adsorption process. Theoretical free energy value (ΔG°) was calculated by Equations (10) and (11):(10)$$K_{d}=\frac{q_{e}}{C_{e}}$$
(11)$$\Delta G^{o}=-{\mathrm{RTLnK}}_{d}$$
where K_d_ refers to distribution coefficient, and R and T are the gas constant and temperature, respectively.

The enthalpy change (ΔH°) and entropy change (ΔS°) values for phosphate were determined from the linear regression curve of 1/T versus LnK_d_ based on the formula of Van’t H of Equation (12):(12)$${\mathrm{LnK}}_{d}=\frac{\Delta S^{o}}{R}-\frac{\Delta H^{o}}{\mathrm{RT}}$$

### 2.5. Bioavailablity of Adsorbed Phosphate

The bioavailablity of adsorbed phosphate was tested according to the method of NaHCO_3_ extraction [31]. First, 0.2 g of phosphate adsorbed LDHs-BC4 was added to 50 mL 

NaHCO_3_ (0.5 M) and shaken for 30 min (150 r/min, 25 °C). Then, 10 mL of transparent solution was taken, and 0.5 M HCl was put into the solution, and finally chromogenic agent was added and phosphate concentration was measured at 824 nm.

In order to determine the release ability of the adsorbed phosphate, 50 mL DI water was added into a conical flask containing 0.2 g phosphate-adsorbed LDHs-BC4. Every 24 h the solution containing phosphate was filtrated and its content was determined. Then 50 mL fresh DI water was added to the conical flask and the above procedure was repeated.

### 2.6. Pot Experiments

Seed treatment experiments of mung bean refer to the method [32]. In short, different dosages of phosphate-adsorbed LDHs-BC4 were added into a plastic pot. The soil without phosphate-adsorbed LDHs-BC4 was used as a blank control. The budding mung bean seeds were put into each pot and cultured in a constant temperature climate chamber (30 °C, 12 h of light). The plant height and root length of mung beans were measured after 8 days.

## 3. Results and Discussion

### 3.1. Characterization of Samples

#### 3.1.1. SEM

Figure 1 presents the surface morphology of the BC-, LDHs-BC4-, and LDHs-BC4-adsorbed phosphate. It showed that the BC surface was smooth, but some small bumps were relatively evenly distributed on the surface, which was caused by the minerals in the pyrolyzed biomass [33]. In addition, the pyrolyzed BC has a layered structure and provided a lot of space for loading Mg and Fe. After the BC was modified by Mg and Fe, the metal oxides were enriched on the surface of the BC, showing an irregular bulk structure. It might be due to the shrinkage and buckling of metal hydroxides through dehydration and dehydroxylation during pyrolysis [34]. It could be confirmed by the presence of element mapping that co-existing positions of Mg and Fe were the formation of LDHs on BC. SEM-EDS elemental mapping images of the LDHs-BC4 adsorption phosphate showed a large number of crystal structures appeared on the LDHs-BC4 surface (Figure 1c), probably due to the emergence of new phosphate crystals, and phosphate was evenly distributed on the surface of LDHs-BC4. It indicated that surface adsorption could be one of the main mechanisms for phosphate adsorption by LDHs-BC4 [34]. Figure 1c showed that there were many large pores in the LDHs-BC4, and phosphate could enter into the material through intragranular diffusion.

#### 3.1.2. XRD

Figure 2 shows the XRD patterns of BC and LDHs-BCx. The peak values of BC were 22.6° and 29.39°, which were typical silica characteristic peaks, and no other peaks appeared due to the amorphous structure of BC. The peaks at 27.16°, 31.74°, 36.74°, 42.71°, 45.41°, 56.45°, 61.88°, 74.21°, and 78.12° in the LDHs-BCx belonged to the typical peaks of Fe_2_O_3_ and MgO [35]. With the increase in Mg/Fe ratio, the peak strength of Fe_2_O_3_ decreased, while that of MgO increased. In addition, all the composites exhibited a good LDHs reflection peak, implying that the composites had good crystallinity.

#### 3.1.3. N_2_ Adsorption-Desorption

Figure 3a shows that all of the adsorption isotherms of three composite materials belonged to type IV, indicating that all the LDHs-BCx were mesoporous. It could be seen from Table 1, compared with BC, the specific surface area of LDHs-BCx decreased and the average pore diameter increased. As the molar ratio of Mg/Fe increased from 2:1 to 4:1, the specific surface area of the composite increased and the average pore diameter decreased, which might be the reason that many metal particles entered into BC and caused blockage [9,36]. When the Mg/Fe ratio was 4, the surface area was the maximum (194.10 m^2^/g) among the three composites, which might be beneficial to the surface adsorption for phosphate.

#### 3.1.4. FT-IR

In order to further investigate the adsorption mechanism, the FT-IR of LDHs-BC4 before and after adsorption phosphate were analyzed. The result is shown in Figure 4. The peak of the LDHs-BC4 at 556 cm^-1^ belongs to lattice vibration of M-O, indicating that LDHs and BC have been successfully recombined [37]. Moreover, after adsorption, new peaks appeared at 860 cm^-1^ and 1392 cm^-1^, which belong to the tensile vibration of the P-O bond. It indicated that the phosphate was successfully adsorbed to the surface of LDHs-BC4 [38]. The peaks at 3440 cm^-1^ and 1618 cm^-1^ were ascribed to the vibrations of -OH and H_2_O. After adsorption, the corresponding peak intensity of-OH decreased, indicating that -OH might be exchanged with phosphate in the adsorption process [39].

### 3.2. Adsorption Experiments

#### 3.2.1. Effect of Different Materials on Phosphate Removal

The effect of different materials (BC and LDHs-BCx) on phosphate adsorption are exhibited in Figure 5. The adsorption efficiency of LDHs-BC2, LDHs-BC3, and LDHs-BC4 was 34.3%, 48.1%, and 81.8%, respectively, and the recovery rate was about 4–10 times higher than that of BC. Among the three composites, the adsorption efficiency of LDHs-BC4 was the highest, which might be related to its larger specific surface area and porosity, and richer special functional groups. However, although BC had the highest specific surface area, it showed the lowest adsorption efficiency for phosphate (only 7.2%) due to the lack of functional groups and smaller pore size.

#### 3.2.2. Influence of pH

Figure 6 illustrates the phosphate adsorption by LDHs-BC4 at different pH conditions. The adsorption efficiency of LDH-BC4 decreased with the increase in pH. Some research reported that the zeta potential of LDHs-BC was about 6 -11 [24,32,34,35], indicating that the -OH on the surface of the adsorbent was protonated to -OH_2_^+^ under acidic conditions (Equation (13)), making it easier for negatively charged phosphate to bind to the surface, leading to the increasing of adsorption efficiency at pH < 7 [37,40]. With the increasing of the pH, the surface negative charge of LDHs-BC4 increased, resulting in a larger electrostatic repulsive force between phosphate and LDHs-BC4. In addition, hydroxides in solution might reduce the effective adsorption sites of LDHs-BC4 by participating in competitive adsorption leading to its lower adsorption efficiency at higher pH.
(13)$$\mathrm{Acidic}\;\mathrm{pH}:\;M-\mathrm{OH}+H^{+}\to M-{\mathrm{OH}}_{2}^{+}$$

#### 3.2.3. Effect of Coexisting Anions

In the real environment, there were many anions (e.g., Cl^−^, SO_4_^2−^, CO_3_^2−^) that coexisted and competed with phosphate for adsorption sites, which might affect the adsorption efficiency of LDHs-BC4 for phosphate. Thus, the effect of co-existing anions on phosphate removal was studied. Figure 7 shows that Cl^−^, CO_3_^2−^, and SO_4_^2−^ had negative effects on phosphate removal, which might be due to competitive adsorption between these anions and phosphate [22,26].

#### 3.2.4. Adsorption Kinetics

The adsorption kinetics fitting curve and data by different kinetic models are displayed in Figure 8 and Table 2. The adsorption capacity increased rapidly at the first 5 h and almost reached saturation at 18 h. The findings demonstrated that the kinetic data had a superior fit with the Elovich model (R^2^ = 0.99). Moreover, the value of α was higher than β, indicating that the phosphate adsorption mainly depended on the early surface adsorption [41]. In addition, the high R^2^ values (R^2^ > 0.8) of the pseudo-second-order kinetic model suggested that chemisorption played an important role in adsorption process. Figure 8b shows that the adsorption process has three well-fitted stages (R^2^ > 0.9). Phosphate diffused swiftly on the surface of LDHs-BC4 with a high K_p_ (K_p1_ = 0.62) value, according to the results of the first stage of adsorption. At the second stage, phosphate gradually diffuses into the material because the active site on the surface of LDHs-BC4 is gradually occupied [42]. The k_p_ value (K_p2_ = 0.18) in this stage was lower than that in the first stage, indicating that intraparticle diffusion process limited the adsorption rate [43]. From the fitting of intramolecular diffusion model, it could be seen that the phosphate adsorption process mainly occurs on the surface of LDHs-BC4.

#### 3.2.5. Adsorption Isotherms

The adsorption isotherms fitted curves are presented in Figure 9. The phosphate adsorption capacity of LDHs-BC4 steadily increased with the increase in initial phosphate concentration. The maximum adsorption of LDHs-BC4 was 10.64 mg-P/g. The phosphate adsorption capacity of LDHs-BC4 exceeded or was comparable with other similar adsorbents reported in literatures (Table 3). Because of the low-cost and perfect adsorption performance of LDHs-BC4, it is a promising phosphate adsorbent. It could be seen from the isothermal fitting data in Table 4 that all models could better fit the isotherm data. (R^2^ > 0.9). The Langmuir model implied that the phosphate adsorption on surfaces of LDHs-BC4 occurred by homogeneous via monolayer [23]. Additionally, the Langmuir–Freundlich model hypothesized that both physical and chemical processes were involved in adsorption. [30], which was coincident with the kinetics. 

#### 3.2.6. Adsorption Thermodynamic

The thermodynamic parameters for phosphate adsorption are shown in Table 5 and Figure 10. Table 5 shows that the G^0^ values were all negative at various temperatures, demonstrating the spontaneity of the adsorption process. Additionally, the G^0^ value dropped with the increasing of temperature, indicating that adsorption was more advantageous at a higher temperature [23]. This was also confirmed by the positive values of ΔH^0^, suggesting the adsorption was an endothermal process. The positive value of ΔS^0^ indicated the LDHs-BC4 had strong affinities with phosphate [30]. 

#### 3.2.7. Adsorption Mechanism

XPS analysis was used to measure the chemical composition and electron valence changes of LDHs-BC4 before and after adsorption, and the result is shown in Figure 11. Figure 11a shows that there were C, O, Fe and Mg in the LDHs-BC4. A peak of P 2p appeared at the binding energy of 133.98 eV after adsorption, and the successful phosphate adsorption was further confirmed. The spectrum of P 2p (Figure 11b) could be decomposed into two peaks of 134.76 and 133.74 eV, belonging to HPO_4_^2−^ and PO_4_^3−^, respectively [22]. The O 1s might be deconvoluted into three peaks with M-O, O-C = O, and M-OH associated with binding energies of 530.6, 533.2, and 532.18 eV, respectively [9,47]. After adsorption, the area of M-OH decreased, indicating that ligand exchange occurred during the adsorption process [22], which was consistent with the FT-IR results. 

Based on the above analysis, predominant adsorption mechanisms of LDHs-BC4 for phosphate included electrostatic attraction, ion exchange, ligand exchange, and intragranular diffusion, which could be elucidated in Figure 12.

Phosphate was firstly adsorbed on the surface of LDHs-BC and then entered the LDHs-BC4 through intramolecular diffusion [22]. The H^+^ at acidic solution protonated the LDHs-BC4 surface and the-OH^2+^ groups were beneficial to adsorb phosphate by electrostatic attraction [48]. Meanwhile, chlorine ions in the layers of the LDHs-BC4 could be exchanged with phosphate [49]. At the same time, hydroxyl group in the LDHs-BC4 could also be replaced by phosphate through ligand exchange. These coordination mechanisms were the key for LDHs-BC4 to efficiently recycle phosphate from wastewater.

#### 3.2.8. Phosphate Release and Bioavailability

The extractable phosphate in the LDHs-BC4-adsorbed phosphate is about 0.785 mg-P/g, which is much higher than the optimal phosphate of 0.045–0.050 mg-P/g in soil for plant growth and crop yield [50]. This result indicates the high feasibility of using LDHs-BC4 to adsorb phosphate from wastewater and then apply it directly to soil as a possible fertilizer.

In order to further investigate the release of P by the phosphate-adsorbed LDHs-BC4, P release amount of the phosphate-adsorbed LDHs-BC4 was measured in the successive 7 days and the results are shown in Figure 13. Except for the highest P release (1.713 mg-P/g) on the first day, the phosphorus release amount in the following 6 days was in the range of 0.22–0.62 mg-P/g (accounting for 1.22–3.55% of the total phosphate). It indicated that the LDHs-BC4 could also be utilized as a possible slow-P-release fertilizer.

### 3.3. Pot Trial

To further test whether recycled phosphate could be applied to soil to prompt crop growth, the effect of different amounts of phosphate-adsorbed LDHs-BC4 on the growth of mung beans was examined. The results are shown in Figure 14. Compared with the control group, when the amount of phosphate-adsorbed LDHs-BC4 was 1% and 10%, the root length and plant height of mung bean showed no obvious significant differences (*p* > 0.05). However, when the dosage of phosphate-adsorbed LDHs-BC4 was 5%, the shoot length of mung bean increased by 36%. There was an obvious significant difference between the experimental group and the control group (*p* < 0.05). It indicated that appropriate dosage of phosphate-adsorbed LDHs-BC4 was beneficial to the growth of mung bean, and too much or too little phosphate-adsorbed LDHs-BC4 was not conductive to crop growth.

## 4. Conclusions

A novel material of LDHs-BC4 was successfully synthesized based on waste jute stalk and used to recover phosphate from wastewater. LDHs significantly improved the structure of BC and its adsorption performance for phosphate. The as-synthesized LDHs-BC4 has significantly high adsorption performance and the recovery rate of phosphate is about 10 times higher than that of the pristine BC. The main phosphate adsorption mechanisms included surface adsorption, electrostatic attraction, ion exchange, ligand exchange, and intragranular diffusion. Moreover, the exhaust LDHs-BC4 could improve the germination of mung bean, which indicated that phosphate-adsorbed LDHs-BC4 could be used as a fertilizer to promote the growth of crops. 

## Figures and Tables

**Figure 1 ijerph-20-03051-f001:**
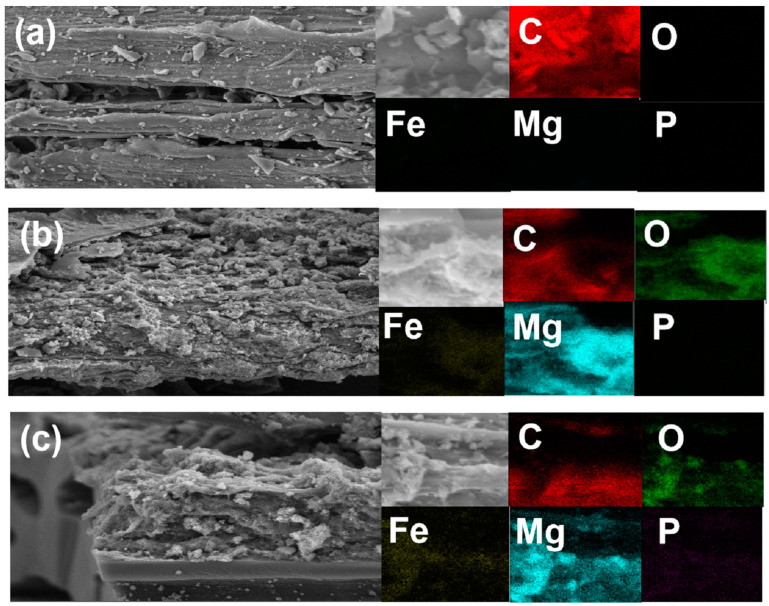
SEM and EDS element mapping of BC (**a**), LDHs-BC4 (**b**), and LDHs-BC4 after phosphate recovery (**c**).

**Figure 2 ijerph-20-03051-f002:**
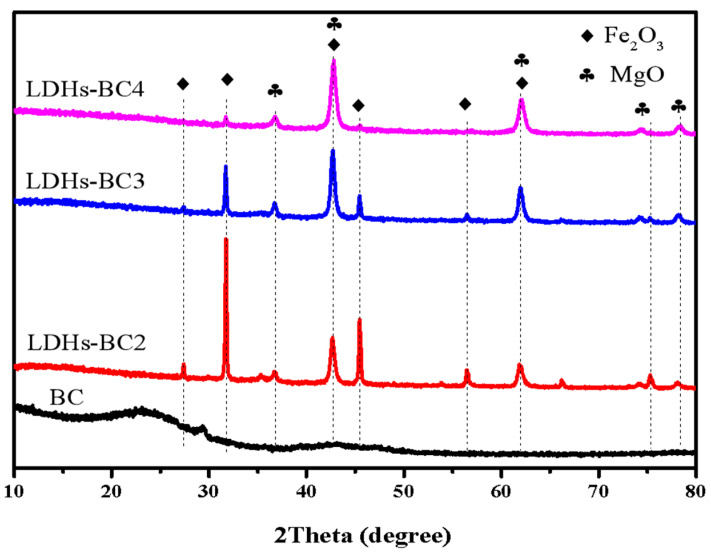
XRD patterns of BC, LDHs-BC2, LDHs-BC3, and LDHs-BC4.

**Figure 3 ijerph-20-03051-f003:**
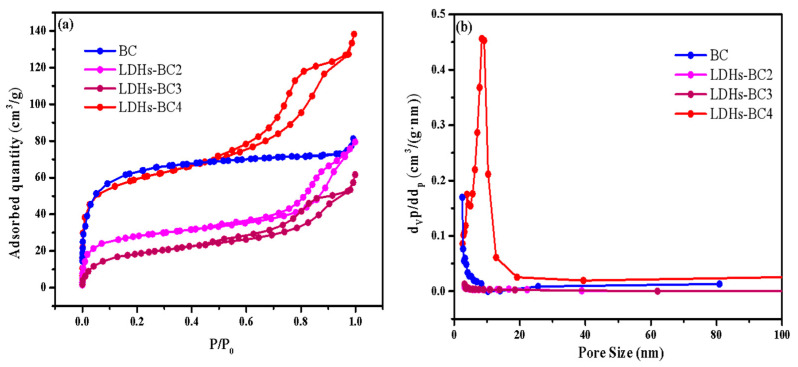
The specific surface area (**a**) and pore size (**b**) of BC and LDHs-BCx.

**Figure 4 ijerph-20-03051-f004:**
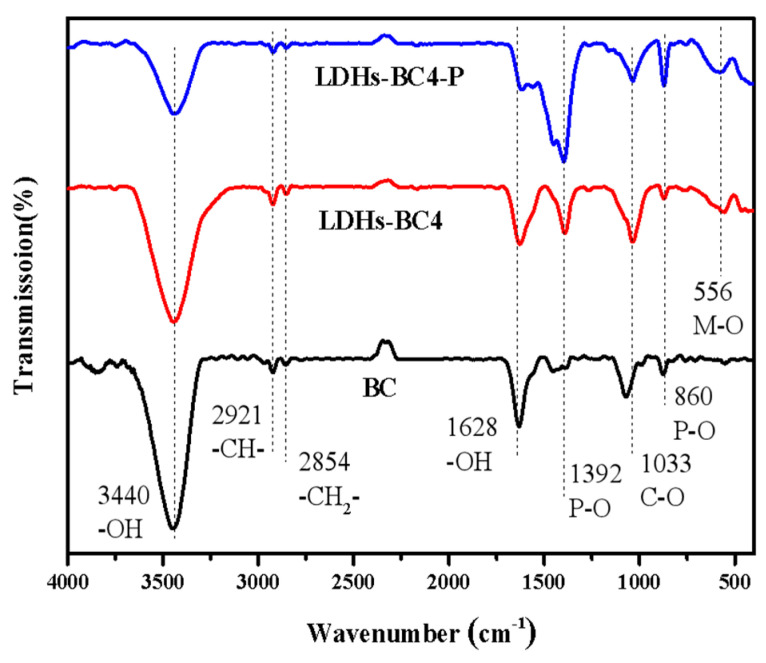
FT-IR spectra of BC, LDHs-BC4, and LDHs-BC4 after phosphate adsorption.

**Figure 5 ijerph-20-03051-f005:**
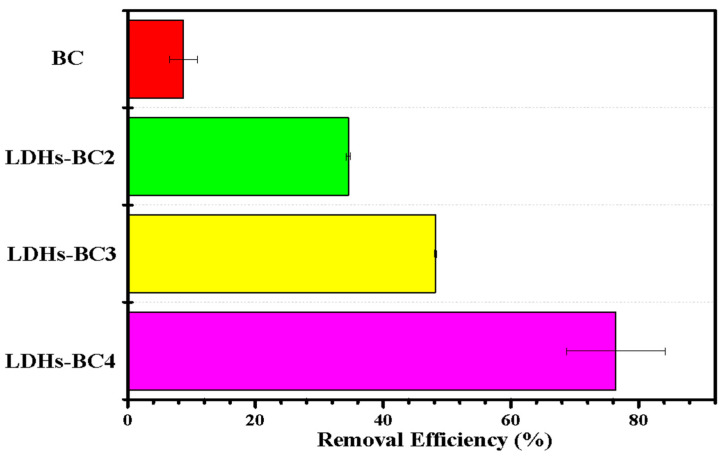
Effect of different materials on phosphate removal.

**Figure 6 ijerph-20-03051-f006:**
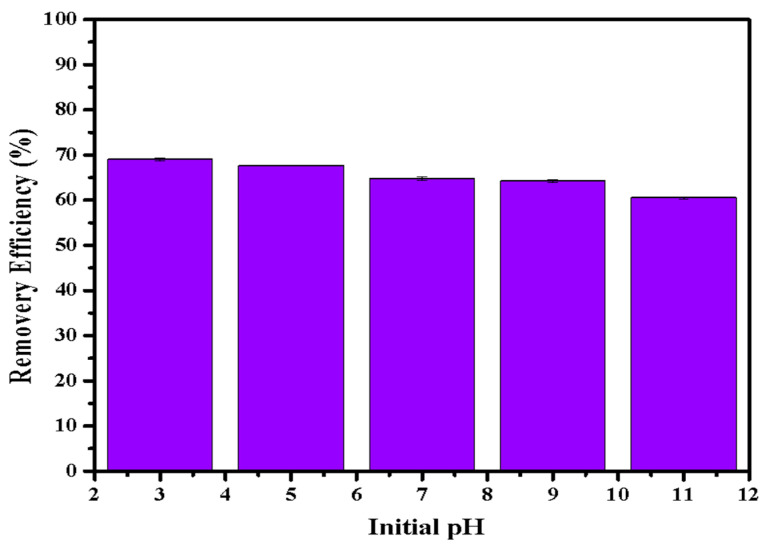
Effect of pH on phosphate removal efficiency.

**Figure 7 ijerph-20-03051-f007:**
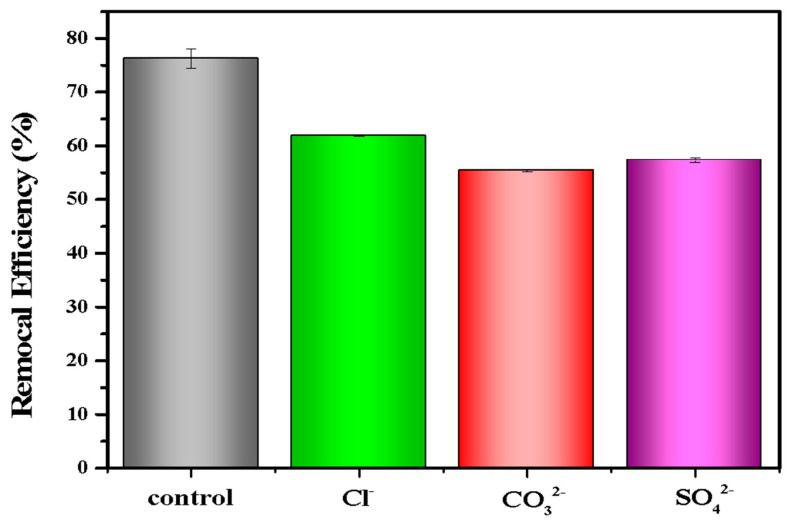
Effect of anions on phosphate recovery efficiency.

**Figure 8 ijerph-20-03051-f008:**
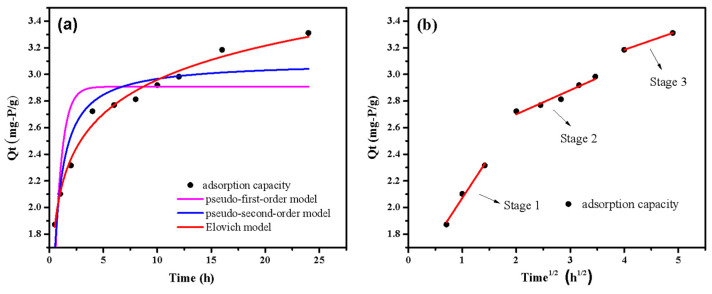
Adsorption kinetics fitting curves by pseudo-first-order kinetics, pseudo-second-order kinetics and Elovich model (**a**) and intraparticle diffusion model (**b**).

**Figure 9 ijerph-20-03051-f009:**
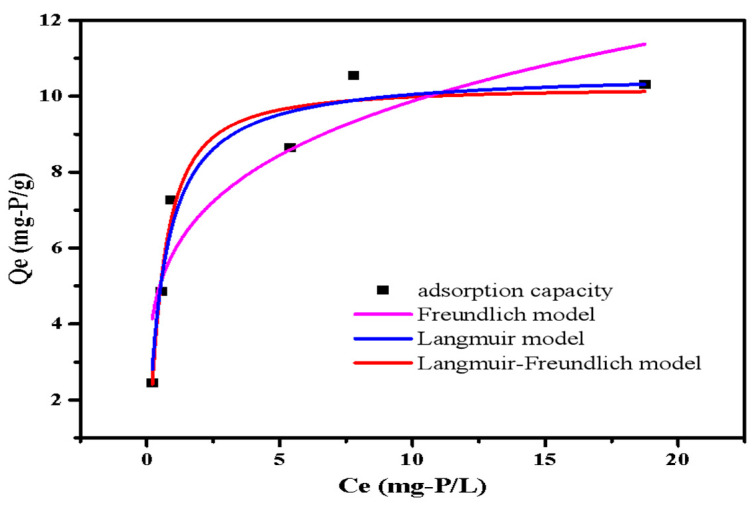
Isotherm models of phosphate adsorption.

**Figure 10 ijerph-20-03051-f010:**
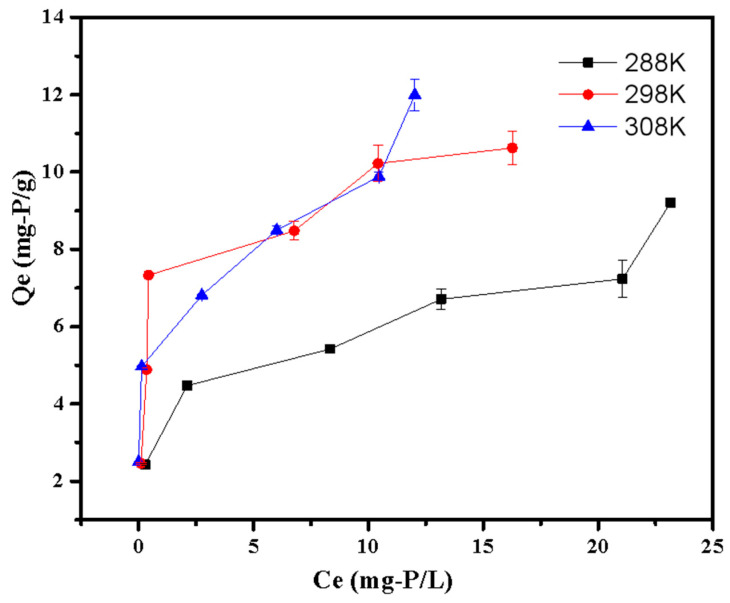
Adsorption thermodynamic of phosphate adsorption.

**Figure 11 ijerph-20-03051-f011:**
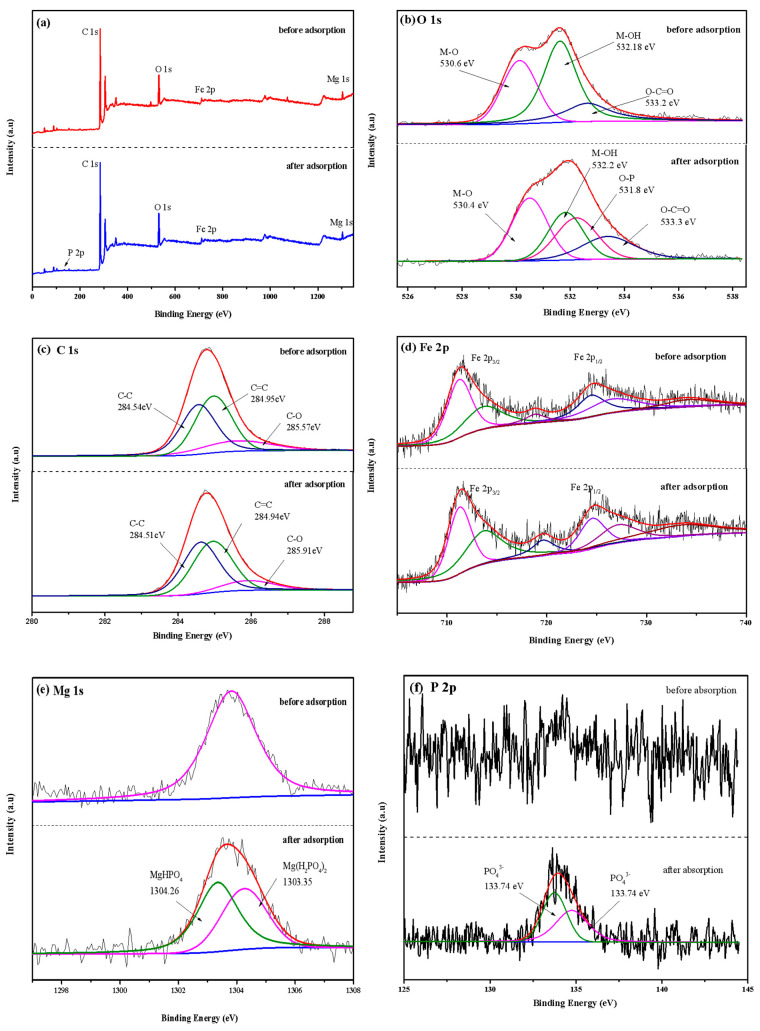
XPS survey spectra (**a**), O 1s (**b**), C 1s (**c**), Fe 2p (**d**), Mg 1s (**e**), and P 2p (**f**).

**Figure 12 ijerph-20-03051-f012:**
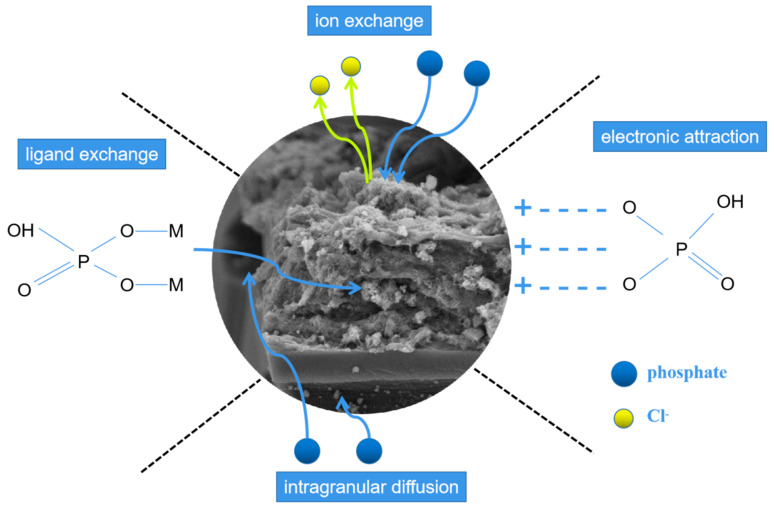
Possible mechanisms of phosphate adsorption.

**Figure 13 ijerph-20-03051-f013:**
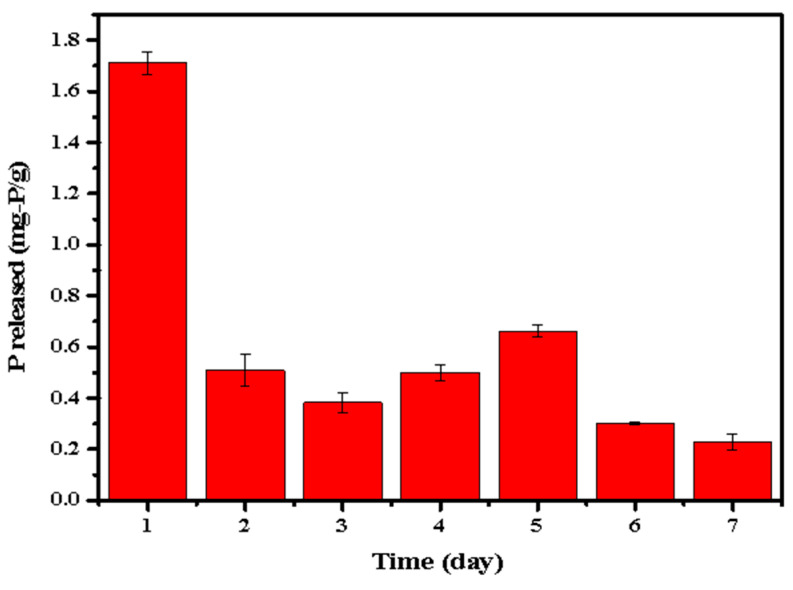
Successive and repeatable release of phosphate.

**Figure 14 ijerph-20-03051-f014:**
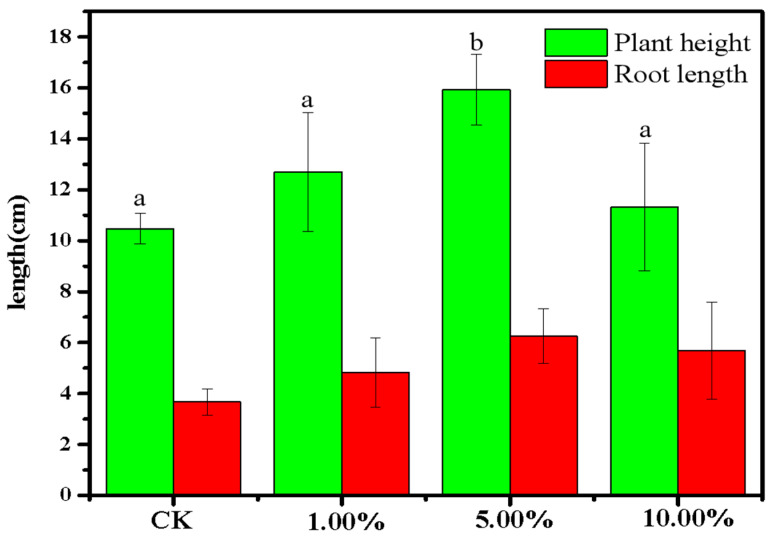
The growth of mung bean treated with phosphate-adsorbed LDHs-BC4. Different letters indicate statistically significant differences between the groups (*p* < 0.05).

**Table 1 ijerph-20-03051-t001:** Specific surface area and average pore size of BC and LDH-BCx.

Characteristics	BC	LDH-BC2	LDH-BC3	LDH-BC4
Specific surface are (m^2^/g)	209.70	64.54	91.46	194.10
Average pore size (nm)	1.20	2.95	2.69	2.20

**Table 2 ijerph-20-03051-t002:** Calculation parameters of kinetics model of phosphate recovery.

Models	Parameter 1	Parameter 2	R^2^
Pseudo-first-order model	K_1_ = 1.49	Qe = 2.91	0.63
Pseudo-second-order model	K_2_ = 0.73	Qe = 3.10	0.86
Elovich model	a = 111.79	b_1_ = 2.71	0.99
Intraparticle diffusion model	k_p1_ = 0.62	b_1_ = 1.43	0.97
	k_p2_ = 0.18	b_2_ = 2.34	0.93
	k_p3_ = 0.14	b_3_ = 0.62	

**Table 3 ijerph-20-03051-t003:** Comparison of phosphate adsorption capacity.

Absorbent	Temperature (°C)	pH	Max Co (mg/L)	Qm (mg-P/g)	Reference
Mg-Al-LDHs-modified almond shell BC	30	6.5	200	1.99	[18]
Mg-Fe-Cl-LDHs	25	7	40	9.80	[19]
Cu-Al-LDHs-modified sisal carbon fiber	15	8	1500	34.38	[25]
Mg-Fe-LDHs-modified pine cone flake BC	25	4	300	5.70	[26]
Fe-Mg-Mn-LDHs	25	6.5	100	11.20	[37]
Mg-Al-LDHs-modified apple wood BC	22	4	200	18.14	[44]
Zn-Al-LDHs	25	6–9	40	22.32	[45]
Ca-Fe paper mill sludge BC	25	11	60	5.66	[46]
Mg-Fe-modified Jute BC	25	7	50	10.64	This study

**Table 4 ijerph-20-03051-t004:** Calculation parameters of isotherm model of phosphate adsorption.

	Parameter	Date
Langmuir model	Qm (mg-P/g)	10.64
	K(L/mg)	1.69
	R^2^	0.94
Freundlich model	1/n	4.44
	K ((L·mg^−1^) 1/n·mg·g^−1^)	5.88
	R^2^	0.81
Langmuir–Freundlich model	Qm (mg-P/g)	10.25
	K (L/mg)	2.15
	1/n	0.80
	R^2^	0.93

**Table 5 ijerph-20-03051-t005:** Thermodynamic constants of phosphate adsorption.

T (K)	K_d_	ΔG (kJ/mol)	ΔH (kJ/mol)	ΔS (J/mol/K)
288	2.80	−2.46	2.11	81.47
298	3.38	−3.02		
308	4.96	−4.10		

## Data Availability

Not applicable.

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
