# Peer review of "Efficiency Recycling and Utilization of Phosphate from Wastewater Using LDHs-Modified Biochar"

_ijerph, 2023, doi:10.3390/ijerph20043051_

Round 1

Reviewer 1 Report

The manuscript entitled "Efficiency recycling and utilization of phosphate from wastewater using LDHs modified biochar " reported the series of biochar composites (LDHs-BCx) with different molar ratio of Mg/Fe and waste jute stalk for recycling phosphate from wastewater. Based on the quality and novelty of this work, it can be published in the International Journal of Environmental Research and Public

Health after revision. Some specific comments are as follows:

1. The abstract of this manuscript should be revised to present the novelty of this study clear.

2. Full names of LDHs, BC, LDHs-BC and LDHs-BCx should be given when they appear at the first time.

3. More details about the information about the characterization of materials should be provided.

4. Did you do parallel experiments? If so, please add error bars in Fig.5-Fig.10, and Fig.13.

5. The spaces between words or word and interpunction were missing in some places. English usage and grammar should be carefully checked and polished throughout the manuscript.

Reviewer 2 Report

The authors of this manuscript report the fabrication/characterization of biochar modified by LDH with various Mg/Fe ratios. Characterization included the use of varied instrumental methods (SEM, EDS, XRD, XPS, N2 adsoprtion/desoprtion, FTIR) allowing to collect useful pieces of structural information. Adsorption experiments, effect of pH/anions, etc. and kinetic studies were also conducted. Importantly, their material was demonstrated as a slow-release fertilizer. The results are interesting and publication – after significant improvements – may be recommended. However, significant improvements are needed.

1. Introduction, line 32: “Currently, phosphate recycling methods… electrodialysis and so on [4].”

The authors state „and so on” and then give a single reference using FeMgMn layered double hydroxides? The rework of this paragraph is needed to give a logical summary of cited references focusing on ref 6 (a review).

2. p.4, lines 165/166: „…which indicated that surface adsorption could be one of the main mechanisms for phosphate adsorption by LDHs-BC4 [29]” In ref 29, a surface complexation model is mentioned rather than adsorption.

3. p.8, lines 218/219: “LDHs-BC4 had the highest adsorption efficiency for phosphate, which might ascribe to its larger specific surface area” (see Fig. 5). According to Table 1, BC has the highest surface area but it has the lowest adsorption efficiency. And what about the pore size? See p.7, line 189: „…the average pore size decreased(table 1)” Not true - it increases then decreases (LHCD-BC2 has the largest value of 2.95 nm)

4. In a literature search in Scopus (phosphate/LHD/removal from 2018) gives 139 results.

It would be highly useful to insert a table at the end of the manuscript with selected literature examples in comparison with the results of this ms.

5. A major characteristic is superficial manuscript writing

1. Examples only considering p.2

- line 72: …into a tubular furnace and treated at 600… (corrected version)

- ref. 5: correct page numbers are 846–853.

- line 69: Adjust pH of the mixed…correct form: “The pH of the mixed solution was adjusted…”

- line 89: temperatures (15-35 oC) a space is needed before oC also 25 oC (line 92)

2. English usage – examples only from p.2

- line 49: …different molar ratios (corrected)

- line 53: “The influencing factors and adsorption mechanism of phosphate adsorbed by LDHs-BCx.” Where is the predicate?

- lines 55–57: “Finally, we tested the possibility of the adsorbed phosphate to be used as an agricultural phosphate fertilizer.” (corrected)

Seek the help of a professional!

Reviewer 3 Report

Reviewer comments:

This paper reported Efficiency recycling and utilization of phosphate from wastewater using LDHs modified biochar. After carefully review, many experiments have been done, but some issues still need to be addressed before publication.

Special comments:

1.      The Zeta potential of LDHs modified biochar should be added for the pH discussion.

2.      The removal mechanism of P over biochar should be enhanced.

3.      The removal performance of this LDHs modified biochar should be compared with other studies.

4.      How is the reuse performance of this LDHs modified biochar?

5.      How is the effect of ionic strength on P removal?

6.      The figures should be further improved, and Error bar should be added.

5. For the application of biochar for the removal of pollutants like P, the following literatures can help to better understand this manuscript: Journal of Hazardous Materials 416 (2021) 125930; Journal of Environmental Chemical Engineering 10 (2022) 107396; journal of environmental sciences 113 (2022) 231–241.

Round 2

Reviewer 2 Report

I am satisfied with authors’ effort to improve their manuscript as indicated in my evaluation. I suggest publication without any further reservation. A remark: English would have been better; nevertheless, it is acceptable.

Reviewer 3 Report

Accept in present form